# The Impact of Intraoperative Glucagon on the Diagnostic Accuracy of Intraoperative Cholangiogram for the Diagnosis of Choledocholithiasis: Experience from a Large Tertiary Care Center

**DOI:** 10.3390/diagnostics14131405

**Published:** 2024-07-01

**Authors:** Nitish Mittal, Faisal S. Ali, Antonio Pizuorno Machado, Sean Ngo, Malek Shatila, Tomas DaVee, Nirav Thosani, Vaibhav Wadhwa

**Affiliations:** 1Department of Internal Medicine, The University of Texas Health Sciences Center, Houston, TX 77054, USA; nitish.mittal@uth.tmc.edu (N.M.); antonio.pizuornomachado@uth.tmc.edu (A.P.M.); 2Department of Gastroenterology, Hepatology, and Nutrition, The University of Texas Health Sciences Center, Houston, TX 77054, USA; faisal.s.ali@uth.tmc.edu (F.S.A.); roy.t.davee@uth.tmc.edu (T.D.); 3School of Medicine, McGovern Medical School, Houston, TX 77054, USA; sean.b.ngo@uth.tmc.edu; 4Department of Gastroenterology, University of Texas MD Anderson Cancer Center, Houston, TX 77030, USA; mshatila@mdanderson.org; 5Department of Surgery, Section of Endoluminal Surgery and Interventional Gastroenterology, McGovern Medical School at UTHealth, Houston, TX 77054, USA; nirav.thosani@uth.tmc.edu

**Keywords:** intraoperative glucagon, intraoperative cholangiogram, laparoscopic cholecystectomy, choledocholithiasis, endoscopic retrograde cholangiopancreatography, diagnostic accuracy

## Abstract

A proportion of patients who undergo intraoperative cholangiogram (IOC) do not have bile duct stones at the time of endoscopic retrograde cholangiopancreatography (ERCP), either due to the spontaneous passage of stones or a false-positive IOC. Glucagon has been utilized as an inexpensive tool to allow the passage of micro-choledocholithiasis to the duodenum and resolve filling defects caused by stones or air bubbles. The purpose of our study is to understand the change in diagnostic accuracy of IOC to detect choledocholithiasis with intraoperative glucagon. We conducted a retrospective study at a tertiary care center on adult patients who underwent laparoscopic cholecystectomy with IOC. The diagnostic accuracy of IOC was assessed before and after the administration of intravenous glucagon. Of 1455 patients, 374 (25.7%) received intraoperative glucagon, and 103 of these 374 patients (27.5%) showed resolution of the filling defect with the passage of contrast to the duodenum. Pre- and post-glucagon administration comparison showed enhancement in specificity from 78% to 83%, an increase in positive predictive value from 67.3% to 72.4%, and an improvement in the diagnostic accuracy of IOC from 81.5% to 84.3%. Our findings suggest that intraoperative glucagon administration carries the potential to reduce the rate of false-positive IOCs, thereby reducing the performance of unnecessary ERCPs.

## 1. Introduction

Gallstone disease is among the most common surgical pathologies encountered globally. Choledocholithiasis is a consequence of gallstone disease that often necessitates biliary intervention, generally through endoscopic retrograde cholangiopancreatography (ERCP), a procedure that, although of established therapeutic potential, carries a significant morbidity burden. Hence, efforts to avoid unnecessary ERCPs have been a focus of intense study and guidance recommendations over the past two decades. One cost-effective approach to avoid unnecessary ERCPs is to perform upfront cholecystectomy with intraoperative cholangiography (IOC) in patients with intermediate probability for choledocholithiasis; those with IOC findings suggestive of choledocholithiasis subsequently undergo ERCP.

A proportion of the patients that undergo IOC do not have bile duct stones at the time of ERCP, either due to the spontaneous passage of stones or due to a false-positive cholangiogram at the time of cholecystectomy. The sensitivity and specificity of IOC have been reported to be 90% and 80%, which is comparable to the diagnostic accuracy of magnetic resonance cholangiopancreatography (MRCP) and endoscopic ultrasound (EUS) [1,2,3]. However, like most diagnostic tests, IOC is prone to operator error and can lead to false-positive findings. This gap in diagnostic ability can be addressed with the help of adjunctive modalities that improve the diagnostic accuracy of IOC. 

Glucagon is a pharmacological agent that inhibits gastrointestinal and duodenal motility by relaxing smooth muscles, decreasing the frequency and amplitude of phasic activity of the sphincter of Oddi [4,5,6]. This property has enabled endoscopists to perform selective biliary cannulation (SBC) more successfully [5,6]. Dalal et al. showcased that the administration of intravenous glucagon during magnetic resonance cholangiopancreatography (MRCP) improved visualization of the CBD and Ampulla of Vater [7]. Intraoperative glucagon administration is commonly performed by surgeons who are experienced in the execution of IOC as it is thought to improve diagnostic accuracy by allowing the passage of micro-choledocholithiasis and resolving air bubbles that can be falsely interpreted as filling defects. Preliminary evidence suggests that intraoperative glucagon may lead to improvement in the diagnostic ability of IOC. However, the body of evidence is very limited, with no studies reporting on the impact of glucagon administration on individual components of the IOC, namely filling defects, biliary dilation, and flow of biliary contrast into the duodenum. Our study aims to address this knowledge gap.

## 2. Methods

We conducted a retrospective study at a large tertiary care center of adult patients who underwent laparoscopic cholecystectomy with IOC from February 2013 to December 2021. Patients were divided into groups based on the presence of stones in the common bile duct on subsequent ERCP. The selection criteria included patients over 18 years undergoing IOC and/or ERCP for a specific reason. Patients’ demographic characteristics, symptomatology on presentation (nausea, vomiting, abdominal pain), and values of total bilirubin, alanine transaminase (ALT), aspartate transaminase (AST), alkaline phosphatase (ALP) at the time of admission were recorded. 

### Cholangiogram and Glucagon Administration

An intraoperative cholangiogram was interpreted by the surgeon operator. Variables assessed for test performance of IOC included the following: (1) dilation of the common bile duct; (2) identification of a filling defect by the operator; (3) lack of duodenal outflow of contrast (representing a lack of biliary outflow obstruction). The decision to administer glucagon was at the discretion of the operating surgeon. When prompted, a standard dose of 1 mg of intravenous glucagon was administered, and a repeat IOC was performed within 5 min of glucagon administration. The three IOC test characteristics were then reassessed. A post-operative ERCP was considered the reference test for the true detection of choledocholithiasis. Patients who did not require and did not undergo postoperative ERCP were considered to have had a true negative IOC.

The diagnostic accuracy of IOC was calculated by assessing the sensitivity, specificity, positive predictive value (PPV), and negative predictive value (NPV) of the cumulative IOC findings as interpreted by the operating surgeon. To further clarify the impact of glucagon on IOC, we assessed diagnostic accuracy at the level of the three individual test findings described above. Descriptive statistics were employed to summarize the findings. For measures of diagnostic accuracy, descriptive statistics, along with their 95% confidence intervals, are reported.

## 3. Results

A total of 1455 patients underwent IOC. The median age of the research population was 38 (29–49), with a median BMI of 30 (26–36) (Table 1). Most of the patients were female (84.8%). In terms of race, the majority were Hispanic (91.6%), followed by African American (4.9%), Caucasian (2.4%), and Asian (0.6%; Table 1). The most common symptoms noted on admission were abdominal pain (99.6%), followed by nausea (83.7%), and vomiting (80.3%). Notably, 796 (54.7%) of patients were diagnosed with acute cholecystitis, and 364 (25.0%) were diagnosed with gallstone pancreatitis.

Of the 1455 IOCs performed, 40.3% were noted to be positive; lack of contrast flow to the duodenum was noted in 67.6% of the patients; filling defects were noted in 80.2% of patients, and a dilated CBD in 15.5% of patients (Table 2). Intraoperative glucagon was used in 374 cases (25.7%). After glucagon administration, 27.5% of the 374 cases had clearance of the filling defect with contrast passage to the duodenum (Table 2). In total, 642 (44.1%) ERCPs were performed; choledocholithiasis was observed in 76.6% of these cases. 

### Diagnostic Yield of Intraoperative Glucagon 

Intraoperative glucagon administration led to improvement in the specificity (78% to 83%) and PPV (67% to 72%) of IOC (as interpreted by the operatic surgeon) to detect choledocholithiasis (Table 3). After glucagon administration, there was an improvement in the specificity (86% to 89%) and PPV (70% to 73%) of filling defect detection on IOC (Figure 1A,B). The specificity (89% to 93%) and PPV (69% to 76%) also improved for finding a lack of duodenal contrast flow on IOC (Figure 2A,B).

## 4. Discussion

This was a large cohort study of data obtained from a high-volume tertiary care center to provide an empirical basis for the use of intraoperative glucagon while performing IOCs. We found that glucagon particularly impacts the specificity and PPV of filling defect detection and identification of duodenal flow of contrast. Taken together, these findings translate into a potential reduction in false-positive results, thereby avoiding unnecessary ERCPs. 

Glucagon is a pharmacological agent that inhibits gastrointestinal and duodenal motility by relaxing smooth muscles and has a powerful choleretic effect, providing a substantial increase in the bile flow within 10 min of intravenous glucagon [8]. A large prospective double-blinded randomization study showed that a combination of glucagon and nitroglycerin prevents post-ERCP complications such as pancreatitis and cholangitis [9]. Evan et al. demonstrated that 1 mg or 2 mg intravenous glucagon has shown similar effectiveness during cholangiography [10]. However, side effects of glucagon should also be considered while administration, such as hyperkalemia, hypoglycemia, chest pain, and headache. We should have an appropriate catecholamine antagonist present while administering glucagon to prevent cardiac crisis in patients with unsuspected pheochromocytoma [11]. Thus, there has been an ongoing debate on the use of glucagon in the surgical literature, which has led to inconsistent use of glucagon during procedures.

The diagnostic accuracy of IOC with and without intraoperative glucagon use to diagnose choledocholithiasis has been studied in a small number of studies (Table 4). Syed et al. highlight an important concern: although IOC demonstrated high sensitivity and NPV of >90% for ruling out retained choledocholithiasis, specificity was modest, and in 162 patients who received ERCP, 14 developed complications, such as pancreatitis (11), perforation (2), and bleeding (1) [12]. Three out of fourteen patients received diagnostic ERCP for evaluation of false-positive IOC. Post-ERCP acute pancreatitis (PEP) is the most frequent complication, and despite the improvement of endoscopic techniques, the PEP rate remains stable over time [13,14]. Boicean et al. published a study in 2023 discussing predictors of PEP in choledocholithiasis extraction [15]. The study analysis shows that female patients remain an important risk group. In contrast, the association between the Dormia basket and balloon dilation procedures in extracting common bile duct stones showed positive results [15]. Moreover, when multiple risk factors are present—such as being female, undergoing extensive interventions, and having sphincter of Oddi dysfunction—the risk of PEP can increase up to 40% [16]. Additionally, blood tests have been used to predict PEP, but extensive studies have remained limited to amylase and lipase levels [17]. There is a paucity of literature on the risk of post-ERCP pancreatitis stratified by findings of a positive IOC, and further studies are needed to address this knowledge gap. In terms of other post-ERCP complications, post-ERCP bleeding has an incidence of 0.3–9.6% and a mortality rate of 0.04% [18]. Perforation is a rare ERCP-related adverse event with an incidence of 0.08–0.6% and an overall mortality rate of 0.06% [12,14]. Few cases in the literature reported ERCP-associated infected intrahepatic pancreatic pseudocyst [19,20]. Their management usually involves endoscopic or percutaneous drainage, with operative intervention reserved for severe infection or ruptured pseudocyst [19,20]. Hence, given the adverse events involved with ERCP, the impact of intraoperative glucagon on augmenting the diagnostic accuracy of IOC for the diagnosis of choledocholithiasis becomes even more crucial. 

It is essential to realize that all the procedures come with a financial burden. Limited data address the cost-effectiveness of evaluation and management strategies in patients with choledocholithiasis. An extensive modeling study assessed the role of EUS and MRCP in patients at intermediate risk of choledocholithiasis. It showcased a cost-saving strategy by avoiding the expense and adverse events of ERCP and its complications [27,28,29,30]. Buscarini et al. conducted a study of intermediate and high-risk choledocholithiasis patients that compared the cost of EUS before ERCP versus ERCP, and it showed the former was a more cost-effective approach [27]. Similarly, Scheiman et al. compared the cost of MRCP versus EUS versus ERCP for patients at intermediate risk of choledocholithiasis using Medicare reimbursements, which indicated expenses of $407, $680, and $1145, respectively [28]. These studies reflect on the financial burden of performing ERCP for false positive cases. A cost-effectiveness analysis suggested that prophylactic pancreatic stent placement was a cost-effective strategy for the prevention of post-ERCP pancreatitis in high-risk patients but still comes with a cost of ~11,000 [31]. Hence, our study provides an aiding tool, intraoperative glucagon during IOC, to manage patients with choledocholithiasis more effectively by determining whether ERCP is required.

For clinical use, the most valuable role of glucagon involves cases where the CBD does not empty into the duodenum, where the use of glucagon increases the diagnostic accuracy of IOC. The Society of American Gastrointestinal and Endoscopic Surgeons (SAGES) guidelines strongly recommend using the IOC technique as it has a high sensitivity, specificity, PPV, NPV, and accuracy for detecting choledocholithiasis [32]. SAGES also suggest that glucagon may be useful during IOC to allow small stones or contrast to pass into the duodenum by relaxing the sphincter of Oddi [32]. Given its low side effect profile, ease of use, and potential improvement in visualization during IOC, glucagon administration should be considered by surgeons when indicated during IOC to increase the yield to detect choledocholithiasis. One argument against the routine use of IOC has been prolonging the duration of laparoscopic cholecystectomy, ranging from 4.3 to 18 min [32,33,34,35]. SAGES suggest IOC during laparoscopic cholecystectomy minimally prolongs the duration of the procedure [32]. Hence, randomized controlled trials are warranted to provide robust evidence to support the adoption of routine glucagon use to improve the yield of IOC, particularly since it has implications for reducing unnecessarily performed ERCPs. 

## 5. Limitations

We acknowledge some limitations of our study. The retrospective observational nature of our study brings with it the inherent biases and potential confounders typical of this study design. In particular, our assessment of false negatives does not account for asymptomatic retained stones that did not warrant a post-operative ERCP. Additionally, the passage of small stones after administration of glucagon, while highlighting its therapeutic potential, also leads to empirical inaccuracies when assessing negative IOCs; however, the clinical significance of such inaccuracies may be limited. The strengths of our study include its large sample size and the granularity of our approach to assess the diagnostic performance of IOC.

## 6. Conclusions

The body of evidence on the utility of glucagon when performing IOC remains limited. Our study adds significantly to this body of evidence and provides an empirical basis to support the use of glucagon when performing IOC. The study highlights augmentation in the diagnostic accuracy of IOC to detect choledocholithiasis with intraoperative glucagon administration. This impacts the specificity and positive predictive value of filling defect detection and identification of duodenal contrast flow, thereby decreasing false-positive rates of IOCs, potentially avoiding the performance of an unnecessary ERCP. Further validation and extension of our findings to assess the impact of ERCPs averted by improved diagnostic accuracy of IOC should be carried out through formal cost-effectiveness analyses and randomized controlled trials. 

## Figures and Tables

**Figure 1 diagnostics-14-01405-f001:**
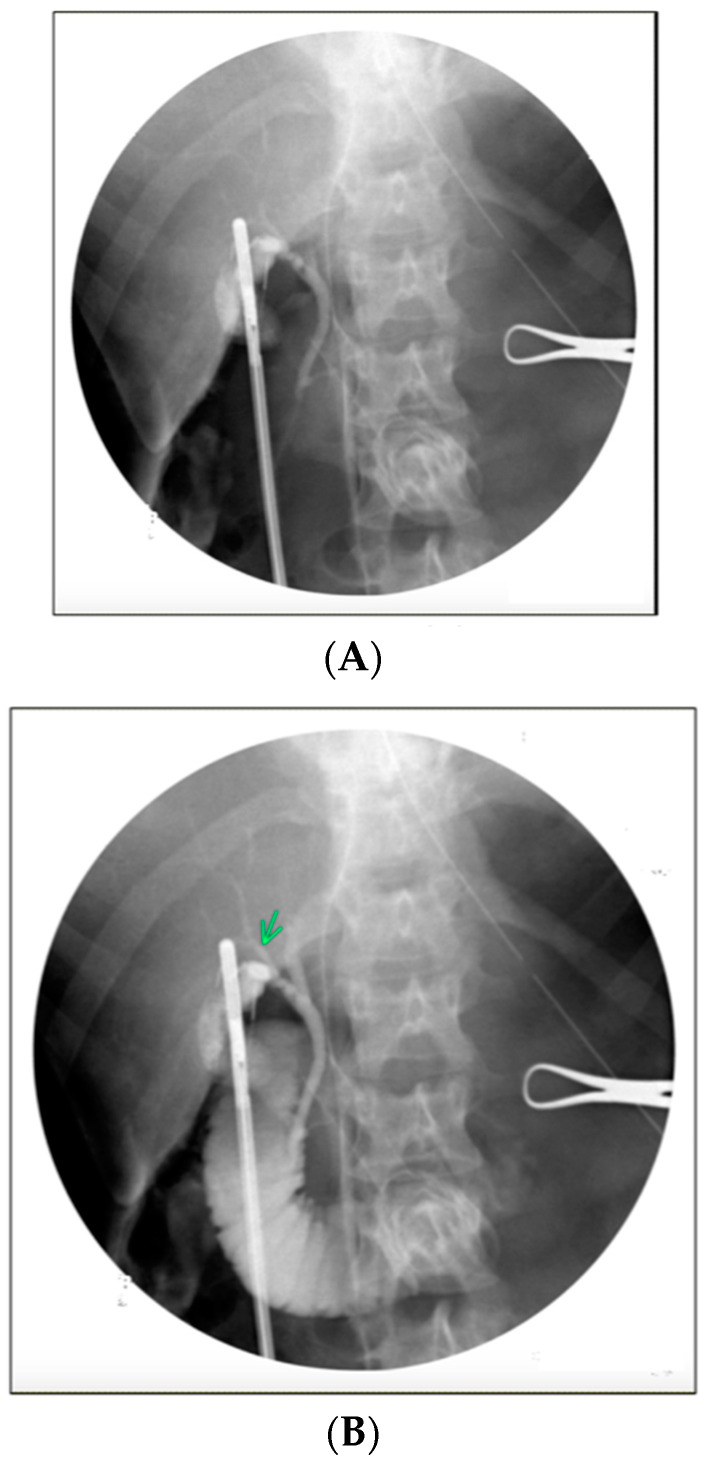
(**A**) Intraoperative cholangiogram with a filling defect. (**B**) Resolution of filling defect after glucagon administration.

**Figure 2 diagnostics-14-01405-f002:**
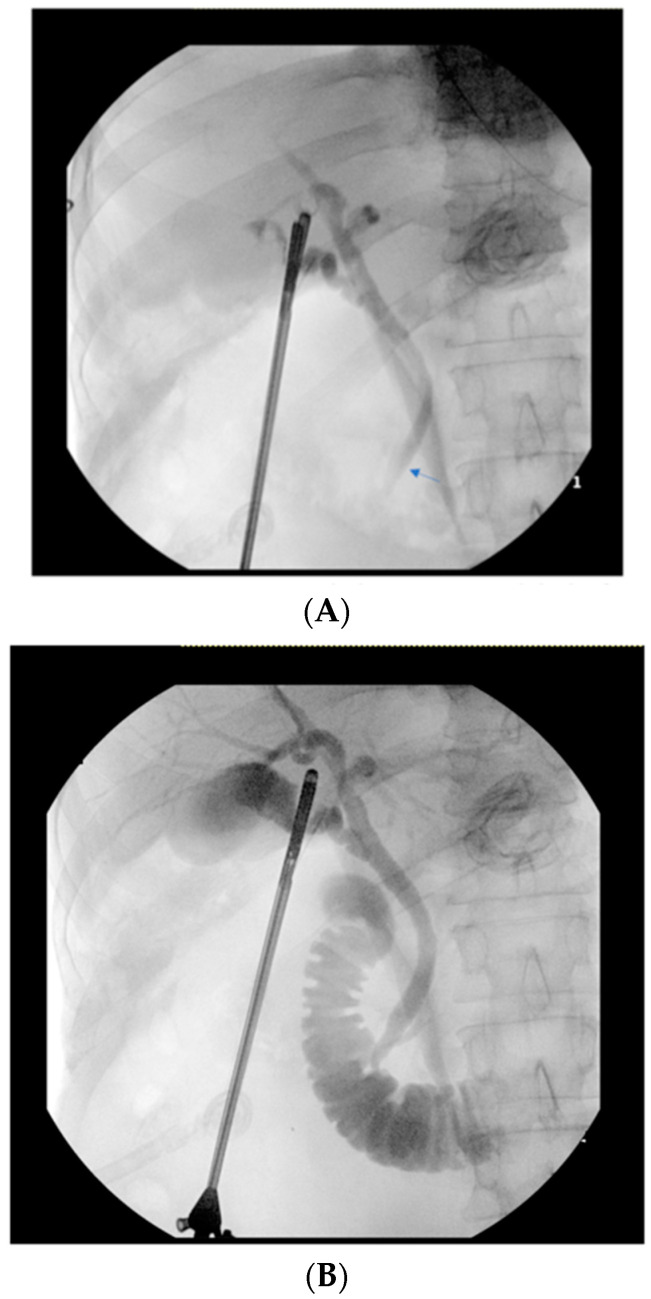
(**A**) Intraoperative cholangiogram with lack of contrast flow into the duodenum from the ampulla. (**B**) Duodenal contrast flow noted after glucagon administration.

**Table 1 diagnostics-14-01405-t001:** Patient characteristics, N = 1455.

Characteristic	No. (%)
Age, years, median (IQR)	38 (29–49)
BMI, median (IQR)	30 (26–36)
BMI = Normal Overweight Obese	259 (17.8) 398 (27.3) 763 (52.4)
Sex = Female Male	1235 (84.8) 220 (15.1)
Race = Hispanic African American Caucasian Asian	1333 (91.6) 72 (4.9) 36 (2.4) 9 (0.6)
Comorbidities	
Hypertension	66 (4.5)
Diabetes Mellitus	63 (4.3)
Hyperlipidemia	32 (2.1)
Smoking (former, current)	42 (2.8)
Alcohol use (former, current)	36 (2.4)
Symptoms on admission Abdominal pain Nausea Vomiting	1450 (99.6) 1218 (83.7) 1169 (80.3)
Cholecystitis	796 (54.7)
Gallstone pancreatitis	364 (25.0)

**Table 2 diagnostics-14-01405-t002:** Operative and ERCP data, N=1455.

Positive IOC	587 (40.3)
No contrast passage to duodenum	397 (67.6)
Filling defects	471 (80.2)
Dilated CBD	91 (15.5)
Intraoperative Glucagon Use	374 (25.7)
Clearance of filling defect with contrast to duodenum Post-Glucagon administration	103 (27.5)
ERCP	642 (44.1)
Bile duct stone on ERCP	492 (76.6)

Abbreviations: ERCP—endoscopic retrograde cholangiopancreatography; IOC—intraoperative cholangiopancreatography; CBD—common bile duct.

**Table 3 diagnostics-14-01405-t003:** Impact of glucagon on the accuracy of intraoperative cholangiogram to diagnose choledocholithiasis.

Category: Intraoperative Cholangiogram (As Interpreted by the Operator)	Pre-Glucagon Administration	Post-Glucagon Administration
Sensitivity (95% CI)	88.3 (85.16% to 91.05%)	86.9 (83.60% to 89.77%)
Specificity (95% CI)	78 (75.33% to 80.68%)	83 (80.48% to 85.35%)
Positive predictive value (95% CI)	67.3 (64.61% to 70.06%)	72.4 (69.42% to 75.19%)
Negative predictive value (95% CI)	92.8 (91.08% to 94.36%)	92.5 (90.77% to 93.97%)
Diagnostic Accuracy	81.5 (79.47% to 83.54%)	84.3 (82.36% to 86.18%)
**Category: Filling Defect on IOC**	**Pre-Glucagon Administration**	**Post-Glucagon Administration**
Sensitivity (95% CI)	83.8 (79.86% to 87.39%)	83.2 (79.20% to 86.86%)
Specificity (95% CI)	86.4 (84.19% to 88.42%)	88.7 (86.64% to 90.56%)
Positive predictive value (95% CI)	69.6 (66.19% to 72.88%)	73.1 (69.56% to 76.43%)
Negative predictive value (95% CI)	93.5 (92.00% to 94.77%)	93.5 (92.01% to 94.73%)
Diagnostic Accuracy	85.7 (83.81% to 87.49%)	87.2 (85.42% to 88.93%)
**Category: Duodenal Contrast Flow**	**Pre-Glucagon Administration**	**Post-Glucagon Administration**
Sensitivity (95% CI)	82.4 (77.88% to 86.38%)	79.5 (74.70% to 83.86%)
Specificity (95% CI)	88.7 (86.75% to 90.56%)	92.9 (91.32% to 94.40%)
Positive predictive value (95% CI)	68.5 (64.68% to 72.11%)	76.2 (71.99% to 79.96%)
Negative predictive value (95% CI)	94.4 (93.08% to 95.56%)	94.1 (92.82% to 95.23%)
Diagnostic Accuracy	87.3 (85.48% to 88.98%)	90 (88.35% to 91.51%)

**Table 4 diagnostics-14-01405-t004:** Literature studies highlighting the impact of glucagon and the diagnostic accuracy of IOC in detecting choledocholithiasis.

Authors	Studies	Findings
Videhult et al. [21]	3-year prospective study in Sweden analyzing benefits of IOC to diagnose choledocholithiasis	Accuracy = 99%; Sensitivity = 97%; Specificity = 99%; NPV = 99%; PPV = 95%
Jamal et al. [22]	Meta-analysis of the diagnostic accuracy of IOC in detection of common bile duct stones	Sensitivity = 0.87; Specificity = 0.98
Ng et al. [23]	5-year tertiary center study explored the incidence and management of choledocholithiasis on routine IOC	474 (13.4%) out of 3904 had positive IOC findings; 158 (33.3%) out of 474 were managed intraoperatively with intravenous glucagon or hyoscine butylbromide
Carroll et al. [24]	A prospective study analyzing feasibility, reliability, and cost of laparoscopic cholangiography versus ERCP	100 patients underwent IOC, 15 had choledocholithiasis on ERCP; 13 out of 15 were suspected of having choledocholithiasis based on IOC, 2 cases were unsuspected
Tabak et al. [25]	A double-blind study of 14 patients explored the glucagon effect on cholangiography	Glucagon provides a statistically significant difference in detecting choledocholithiasis for glucagon cohort
Cofer et al. [26]	A prospective, randomized, double-blind study to test the effects of intravenous glucagon on IOC	Glucagon provides no statistically significant difference for the diagnosis of choledocholithiasis

## Data Availability

The data presented in this study are available on request from the corresponding author due to privacy currently.

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
