# Peer review of "The Impact of Intraoperative Glucagon on the Diagnostic Accuracy of Intraoperative Cholangiogram for the Diagnosis of Choledocholithiasis: Experience from a Large Tertiary Care Center"

_diagnostics, 2024, doi:10.3390/diagnostics14131405_

Round 1

Reviewer 1 Report

Comments and Suggestions for Authors

I congratulate the authors for their great work!

The study really brings to the literature, and the change in clinical practice should be considered even with retrospective design, and as the authors suggested in their conclusion, I am looking forward to large-scale prospective study on this subject.

To my view, one more information would also be important. Since there was a great percentage of patients with lack of contrast out-flow on IOC before glucagon administration (367 pts; 67% of positive IOC), that resolved afterward in 103 or 27%, having in mind the pathophysiology of biliary pancreatitis onset, could more patients develop biliary pancreatitis as this intervention could transitory increase the pressure in main pancreatic duct and/or cause inflow of bile into pancreatic ducts during distal common bile duct stone passage? I assume that this question could also be assessed.

Author Response

Responses to Reviewer 1 Comments

Summary:

Thank you very much for taking the time to review the manuscript. Please find the detailed responses below, and the corresponding revisions in the re-submitted files highlighted in red.

Point-by-point response to Comments and Suggestions for Authors:

Comment 1: The study really brings to the literature, and the change in clinical practice should be considered even with retrospective design, and as the authors suggested in their conclusion, I am looking forward to large-scale prospective study on this subject.

Response 1: Thank you so much for your encouraging words. We are planning for a large-scale prospective study on this subject in future.

Comment 2: To my view, one more information would also be important. Since there was a great percentage of patients with lack of contrast out-flow on IOC before glucagon administration (367 pts; 67% of positive IOC), that resolved afterward in 103 or 27%, having in mind the pathophysiology of biliary pancreatitis onset, could more patients develop biliary pancreatitis as this intervention could transitory increase the pressure in main pancreatic duct and/or cause inflow of bile into pancreatic ducts during distal common bile duct stone passage? I assume that this question could also be assessed.

Response 2: We thank reviewer for the comments. This is a very interesting observation highlighted by the reviewer. We looked at the data base carefully, and we have not seen it clinically causing pancreatitis. So, while this remains physiologically essential, it clinically remains non-significant based on our research data collection. There is a paucity of literature in post-ERCP pancreatitis among patients who did not have duodenal contrast flow on IOC, and further prospective studies need to be performed to establish this correlation. We have added extra data in lines 152-161 with table 4 highlighting studies on the diagnostic accuracy of IOC with and without intraoperative glucagon use to diagnose choledocholithiasis.

Reviewer 2 Report

Comments and Suggestions for Authors

The article The Impact of Intraoperative Glucagon on The Diagnostic Accuracy of Intraoperative Cholangiogram for The Diagnosis of Choledocholithiasis - Experience from a large Tertiary Care Center discusses the impact of intraoperative glucagon on the diagnostic accuracy of intraoperative cholangiogram, a highly sought-after solution in the 21st century. Modifications:

1.       In the Materials and Methods section, add more information about the patient cohort and the selection criteria.

2.       The Results section is very well done.

3.       In the Discussion section, add a summary of the main studies from the literature and their results in a table. Additionally, you can obtain more information about post-ERCP pancreatitis from https://doi.org/10.3390/jpm13091356.

4.       Add a separate section for the limitations of the study.

5.       The Conclusions should be rewritten to better present the findings of the study.

6.       The Bibliography needs improvements – include more recent studies and ensure the formatting meets MDPI standards. For example, in citation 18, the article title should not appear in uppercase.

Author Response

Summary:

Thank you very much for taking the time to review the manuscript. Please find the detailed responses below, and the corresponding revisions in the re-submitted files highlighted in red.

Point-by-point response to Comments and Suggestions for Authors:

Comment 1: In the Materials and Methods section, add more information about the patient cohort and the selection criteria.

Response 1: We thank reviewer for the comments.

Patients were divided into groups based on the presence of stones in the common bile duct on subsequent ERCP. The selection criteria included patients with age greater than 18 years who are undergoing IOC or/and ERCP for a specific reason. Lines: 70-72.

Comment 2: The Results section is very well done.

Response 2: Thank you so much for your encouraging words.

Comment 3:  In the Discussion section, add a summary of the main studies from the literature and their results in a table. Additionally, you can obtain more information about post-ERCP pancreatitis from https://doi.org/10.3390/jpm13091356.

Response 3: Thank you for the insightful comment. We have made the necessary changes with a table for the studies (Table 4 – lines: 172-174), and added more information about post-ERCP pancreatitis (Lines: 152-161).

Table 4. Literature studies highlighting impact of glucagon and the diagnostic accuracy of IOC to detect choledocholithiasis.

Authors

Studies

Findings

Videhult et al. [12]

3-year prospective study in Sweden analyzing benefits of IOC to diagnose choledocholithiasis

Accuracy = 99%, Sensitivity = 97%, Specificity = 99%, NPV = 99%, PPV = 95%

Jamal et al. [13]

Meta-analysis of the diagnostic accuracy of IOC in detection of common bile duct stones

Sensitivity = 0.87, Specificity = 0.98

Ng et al. [14]

5-year tertiary center study explored the incidence and management of choledocholithiasis on routine IOC

474 (13.4%) out of 3904 had positive IOC findings; 158 (33.3%) out of 474 were managed intraoperatively with intravenous glucagon or hyoscine butylbromide

Carroll et al. [15]

A prospective study analyzing feasibility, reliability, and cost of laparoscopic cholangiography versus ERCP

100 patients underwent IOC, 15 had choledocholithiasis on ERCP; 13/15 had suspicion of choledocholithiasis based on IOC, 2 were unsuspected

Tabak et al. [16]

A double-blind study of 14 patients explored the glucagon effect on cholangiography

Glucagon provides statistically significant difference in detecting choledocholithiasis for glucagon cohort

Cofer et al. [17]

A prospective, randomized, double-blind study to test the effects of intravenous glucagon on IOC

Glucagon provides no statistically significant difference for the diagnosis of choledocholithiasis

Comment 4: Add a separate section for the limitations of the study.

Response 4: We created a separate section for the limitations after the discussion and before the conclusion section. Lines: 209-218.

Comment 5: The Conclusions should be rewritten to better present the findings of the study.

Response 5: Thank you for the comment. We have re-structured the conclusion section.

The body of evidence on the utility of glucagon when performing IOC remains limited. Our study adds significantly to this body of evidence and provides an empirical basis to support the use of glucagon when performing IOC. The study highlights augmentation in the diagnostic accuracy of IOC to detect choledocholithiasis with intraoperative glucagon administration. This impacts the specificity and positive predictive value of filling defect detection and identification of duodenal contrast flow, thereby decreasing false positive rates of IOCs, potentially avoiding performance of an unnecessary ERCP. Further validation and extension of our findings to assess the impact of ERCPs averted by improved diagnostic accuracy of IOC should be carried out in formal cost-effectiveness analyses and randomized controlled trials.

Lines: 219-229.

Comment 6: The Bibliography needs improvements – include more recent studies and ensure the formatting meets MDPI standards. For example, in citation 18, the article title should not appear in uppercase.

Response 6: Thank you for the comment. We included more recent studies– references 21-23, and the references have been corrected with the changes highlighted in red.
